# Causal Inference with Non-IID Data using Linear Graphical Models

**Chi Zhang**
Department of CS
UCLA
California, USA, 90095
zccc@cs.ucla.edu

**Karthika Mohan**
Department of EECS
Oregon State University
Oregon, USA, 97331
mohank@oregonstate.edu

**Judea Pearl**
Department of CS
UCLA
California, USA, 90095
judea@cs.ucla.edu

## Abstract

Traditional causal inference techniques assume data are independent and identically distributed (IID) and thus ignores interactions among units. However, a unit's treatment may affect another unit's outcome (interference), a unit's treatment may be correlated with another unit's outcome, or a unit's treatment and outcome may be spuriously correlated through another unit. To capture such nuances, we model the data generating process using causal graphs and conduct a systematic analysis of the bias caused by different types of interactions when computing causal effects. We derive theorems to detect and quantify the interaction bias, and derive conditions under which it is safe to ignore interactions. Put differently, we present conditions under which causal effects can be computed with negligible bias by assuming that samples are IID. Furthermore, we develop a method to eliminate bias in cases where blindly assuming IID is expected to yield a significantly biased estimate. Finally, we test the coverage and performance of our methods through simulations.

## 1   Introduction

**Motivating Example** Suppose we are interested in studying the effectiveness of Covid-19 vaccines. Specifically, we are interested in the causal effect of vaccine doses, $V$, on the severity of sickness $S$. A naive method would be building a causal model on $V$, $S$, and other related factors, and estimating the causal effect of $V$ on $S$ using available data. However, this method may result in biased estimation primarily because traditional causal inference techniques assume that attributes of all units in the sample are independent and identically distributed (IID)[Rubin, 1978], which does not hold true in the pandemic setting since units are not isolated from each other. We exemplify below few instances of this problem that violate IID (Eyre et al. [2022]).

**Case 1:** The vaccination $V$ of a unit $i$, $(V_i)$, decreases their viral load, $L_i$, which in turn decreases the transmission rate of the virus, and hence decreases the severity of sickness $S$ of another unit $j$, $(S_j)$, who comes into contact with $i$. $V_i$ *causally affects* $S_j$.

**Case 2:** $V_i$ is affected by the area $A$ that $i$ lives in, and a contact $j$ who lives in vaccine deprived areas and areas with a higher incidence of Covid-19 infection is more likely to get sick. $V_i$ *and* $S_j$ *are confounded.*

**Case 3:** $S_i$, determines whether or not $i$ is quarantined and thus affects whether $i$ transmits the disease to another unit $j$. $S_i$ *causally affects* $S_j$.

Such interactions between units plague both observational and experimental studies. If the latter is performed in a controlled environment where subjects are isolated from each other, the results would not be valid for the target environment, where subjects affect each other, and vice versa.

**Modeling Interactions** A line of existing work that analyzes interactions between units is interference [Cox, 1958]. Interference is the phenomenon in which treatment of unit $i$ $(V_i)$ causally affects the

outcome, $S_j$, of another unit $j$. In almost all existing literature this is interpreted as there existing a causal pathway from $V_i$ to $S_j$. Case-1 above is a typical example. Clearly, ignoring unit interactions while computing causal effects would result in a biased estimate. However, we note that interference is not the only type of interaction between units that can yield biased estimates. For example, in Case-2 $V_i$ and $S_j$ are confounded and $V_i$ is not a cause of $S_j$. Another example is an instance where unit $i's$ treatment affects their own outcome through an attribute of unit $j$ i.e., $V_i \rightarrow W_j \rightarrow S_i$, for some $W_j \notin \{S_j, V_j\}$. In both these cases units interact with each other in a way that might bias the estimation of causal effects although they may not typically be classified as interference. In spite of the prevalence of such interactions in applications related to health care, infectious diseases, social networks and ad placements, they have not been systematically studied. It is this deficiency that this paper attempts to overcome.

**Questions addressed** The scenario exemplified above raises several questions regarding the computation of causal effects given non-IID data. How can we model different types of interactions among units in the population? Under what conditions can we safely ignore unit interactions with the guarantee that assuming IID (and applying existing estimation techniques) will result in negligible bias? If assuming IID would yield a biased estimate, then how can we get rid of this bias?

**Our contributions** In this work we study causal inference in the presence of interactions among samples using linear models due to the convenience they offer with regard to path analysis. We develop interaction models that portray different types of interactions among units and conduct a systematic analysis of the bias caused by different types of interactions. We derive theorems to detect and quantify the interaction bias. We derive conditions under which it is safe to ignore unit interactions when computing the average causal effects. Furthermore, we develop a method to compute an unbiased estimate of causal effect in cases where blindly assuming IID is expected to yield a significant bias. Finally, we corroborate our findings through simulation studies.

**Summary of results in words** Blindly assuming that data are IID when in fact they are not, can potentially bias the outcome of a research study. Such bias can occur for the query: causal effect of treatment on outcome, when there is an open (not necessarily directed) path from the treatment of unit $i$ to the outcome of unit $j$ and/or to the outcome of unit $i$ itself such that an intermediate node on the path belongs to unit $j$. The formula in Theorem 1 quantifies the bias. Furthermore, only the two types of interaction structures previously mentioned can induce bias. In the presence of such bias inducing structures it is still possible to compute an unbiased estimate by selecting a subset of samples $B$ such that no biasing paths exist in the interaction graph corresponding to samples in $B$ (Theorem 2). More importantly, such a debiasing procedure does not require the selection of IID samples and may contain interactions among them. Such a debiasing procedure can also be done in polynomial time (Algorithm 1). Empirical analysis: We randomly generate interaction models and show that the bias can be huge if IID is wrongly assumed on non-IID data. The debiasing method in this paper yields an unbiased estimate. We further show that, as the number of bias-free samples increases, and as the strengths of bias structures decrease, the overall interaction bias decreases.

## 2   Preliminaries

**Independent and Identically Distributed (IID)** If $X_1, \ldots, X_n$ are independent and each has the same marginal distribution with CDF $F$, we say that $X_1, \ldots, X_n$ are IID (independent and identically distributed) [Wasserman, 2013]. For the sake of simplicity, we use $X$ is IID to refer to all the units of $X, X_1, \ldots, X_n$, being IID. A dataset is IID if all variables in it are IID.

**Linear Causal Models** A traditional linear causal model is also known as a linear structural causal model (SCM) [Brito, 2004, Pearl, 2009, Chen and Pearl, 2014]. The edge coefficients on the causal DAG represent direct effects. An open path is collider-free, i.e., there are no head-to-head arrows on this path. Note that if there exists an open path from $W_i$ to $V_j$, it implies $W_i \not\!\perp\!\!\!\perp V_j$. The value '$Val(p)$' of an open path $p$ in a linear model is defined as the product of the edge coefficients on $p$. The root of an open path $p$ is defined as the variable on $p$ that is the ancestor of all variables on $p$.

**Average Causal Effects** In this work the query we are primarily interested in generalizing to the non-IID case is the Average Causal Effect (ACE), also named as the Average Treatment Effect (ATE) [Rubin, 1977, Holland, 1988]. For consistency, we use ACE to refer to both. Given a causal model $M$, the average causal effect (ACE) of $X = t$ vs $X = c$ ($t$ and $c$ are constants) on $Y$ for $k$ units is defined as $ACE_{XY} = \frac{1}{k} \sum_i (Y_{iX_i=t} - Y_{iX_i=c})$. ACE is defined under the assumption that $Y_i$ depends only

on factors of unit $i$ (including $X_i$) Holland [1988]. Without loss of generality, we assume $t = c + 1$[1]. In linear models, $ACE$ of $X$ on $Y$ can be identified as $\beta_{YX}$, the linear regression slope of $Y$ on $X$, if there is no backdoor (non-directed open paths) between $X$ and $Y$ [Pearl et al., 2016, Pearl, 2017].

## 3 Graphical Modeling of Interactions

### 3.1 Interaction models

In this section, we define a graphical model derived from traditional causal models $M = <G, S>$ (Pearl [2009], definition 7.1.1). $G$ is the causal graph and $S$ is the set of structural equations of variables. We refer to the variables in a traditional causal model as *generic variables*. $X, C, Y$ in Figure 1 are generic variables. An *explicit variable* is similar to a generic variable except that it represents an attribute/event of one specific unit (or sample or individual). For example, "treatment $(X)$" is a generic variable, and "the treatment of unit $i$ $(X_i)$" is an explicit variable.

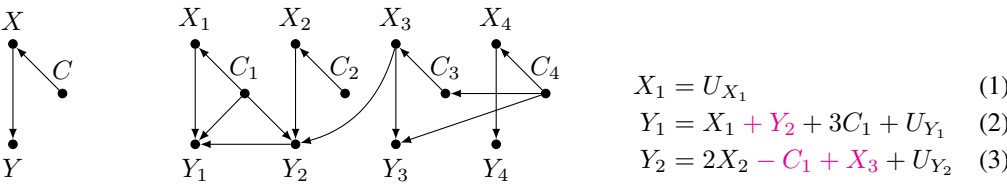

$$X_1 = U_{X_1} \tag{1}$$
$$Y_1 = X_1 + Y_2 + 3C_1 + U_{Y_1} \tag{2}$$
$$Y_2 = 2X_2 - C_1 + X_3 + U_{Y_2} \tag{3}$$
$$\cdots$$

**Figure 1:** Traditional DAG.

**Figure 2:** Interaction network with 4 units and 12 explicit variables ($X_i, Y_i, C_i$ for $i = 1, 2, 3, 4$).

**Definition 1** (Interaction model $M^*(G^*, S^*)$). *An interaction model, $M^*(G^*, S^*)$, is a causal model where $G^*$ is the interaction network and $S^*$ is the set of structural equations defining the data generating process of the observed explicit variables. An interaction network, $G^*$, is a directed acyclic graph with each node representing an explicit variable and each directed edge $A_i \to B_j$ representing $A_i$ causes $B_j$.*

An example of interaction model $M^*(G^*, S^*)$, is the interaction network, $G^*$, portrayed in Figure 2 and the structural equations $S^*$ (part of) specified beside it; $U_{V_i}$ denotes the unobserved exogenous error of an explicit variable $V_i$. Observe that interaction networks allow edges between explicit variables of the same unit (e.g., $X_1 \to Y_1$), as well as two distinct units (e.g., $C_1 \to Y_2$).

We are now ready to define an *isolated interaction model* for an interaction model $M^*$. It is the "ideal" model constructed from $M^*$ by eliminating all interactions between units.

**Definition 2** (Isolated interaction model $IM^*(IG^*, IS^*)$). *$IM^*(IG^*, IS^*)$ is the* Isolated *interaction model of an interaction model $M^*(G^*, S^*)$ if $IM$ satisfies the following conditions:*

1. *$IG^* = G'$ where $G'$ is the graph obtained by removing from $G^*$ all edges $A_i \to B_j$, $i \neq j$,*

2. *$IS^* = S'$ where $S'$ is the set of equations obtained by removing from each equation $X_i = f(Pa(X_i))$[2] in $S^*$ all terms containing any $Y_j$, $\forall j \neq i$.*

For example, the interaction model $M^*(G^*, S^*)$ has Figure 2 as $G^*$, and Equations (1-3) as part of $S^*$. The isolated model for $M^*$ is denoted $IM^*(IG^*, IS^*)$. $IG^*$ is given in Figure 3 below. And $IS^*$ for Equations (1-3) are given by Equations (4-6).

### 3.2 Symmetry Assumptions

In real-world applications, we will have at our disposal limited (usually just one) observations corresponding to a unit which in turn will make it hard to draw useful conclusions if the model is completely arbitrary. In traditional causal inference techniques this is not a problem since they assume IID, which is assuming for each variable, the distribution is the same and independent for

---

[1]If $t \neq c + 1$, the ACE is multiplied by the constant $(t - c)$.

[2]$Pa(X_i)$ denotes the parents of $X_i$ in $G^*$.

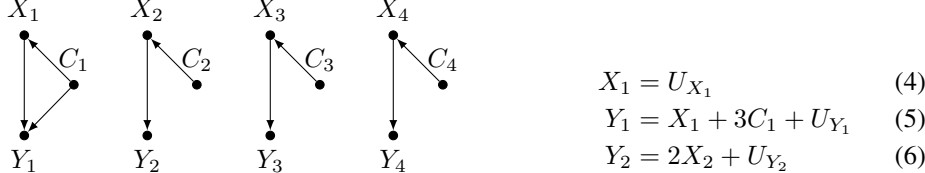

$$X_1 = U_{X_1} \quad (4)$$
$$Y_1 = X_1 + 3C_1 + U_{Y_1} \quad (5)$$
$$Y_2 = 2X_2 + U_{Y_2} \quad (6)$$

**Figure 3:** Interaction network with 4 units.

all units. While we do not make strong assumptions such as IID, we need to make certain weaker symmetry restrictions (definitions 3, 4), in order to quantify bias and identify $ACE$. We only require some of the variables are IID instead of all.

**Definition 3** (Balanced interaction model $M^*(G^*, S^*)$). *Let $M^*(G^*, S^*)$ be an interaction model with isolated model $IM^*$. $M^*$ is a* balanced *interaction model if $IM^*$ has the same unit-model $(IM_i^*(IG_i^*, IS_i^*))$ for every unit $i$.*

Let $G^*$ be the graph in Figure 2 and $S^*$ be the set of equations (1-3) corresponding to $M^*(G^*, S^*)$. $IG^*$ in Figure 3 is the graph and $IS^*$ are the equations (4), (5) and (6) that correspond to $IM^*$, which is the isolated model of $M^*$. The unit-graph for unit 1 is different from unit 2. Also, the structural equations for $Y_1$ and $Y_2$ of the isolated interaction model (Equations (5) and (6)) are different. Hence, $M^*$ is not a balanced interaction model.

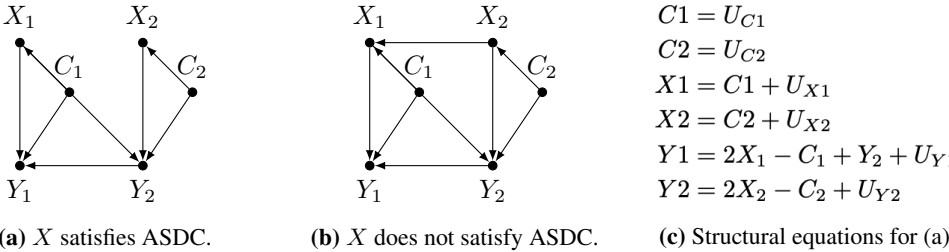

$$C1 = U_{C1}$$
$$C2 = U_{C2}$$
$$X1 = C1 + U_{X1}$$
$$X2 = C2 + U_{X2}$$
$$Y1 = 2X_1 - C1 + Y2 + U_{Y1}$$
$$Y2 = 2X_2 - C2 + U_{Y2}$$

**(a)** $X$ satisfies ASDC.      **(b)** $X$ does not satisfy ASDC.      **(c)** Structural equations for (a)

**Figure 4:** Two balanced interaction networks and the structural equations for (a)

For another example, the interaction model $M^*$ is balanced where $G^*$ is the graph in Figure 4(a), and $S^*$ is the set of equations given in Figure 4(c).

**Remark 1.** *Note that a balanced interaction model $M^*$ does not imply that data generated by it are IID. Being balanced only requires all units share the same causal relationships within each unit itself, but permits interactions and effects from other units. For example, the parents of explicit variables $Y_i$ and $Y_j$, $i \neq j$ can be different in $G^*$ i.e., $Y_i$ can be caused by a set of variables $S_k$ corresponding to unit, $k$, and $Y_j$ can be caused by a distinct set of variables $T_k$. However, for $M^*$ to be balanced it is required that for all distinct units $i$ and $j$, all $Y_i$ have the same relationship with $i$'s explicit variables as $Y_j$ with $j$'s variables.*

We further note that if $M^{**}$ is balanced then all the unit-models $IM_i^*(IG_i^*, IS_i^*)$ in definition 3 are identical (with no edges between $IG_i^*$ and $IG_j^*$), and can be succinctly represented by a (single) causal model $M(G, S)$ where $G$ and $S$ can be constructed from any $IG_i^*$ and $IS_i^*$ by replacing explicit variables with generic variables.

In addition to the assumption that the isolated components being the same, it would be helpful if we also have *symmetrical* assumptions on the underlying distributions of specific sets of variables. For example, it is reasonable to assume all units' treatments have the same distribution, i.e., for any treatment $X = x$, all units have an equal chance of getting the treatment $X = x$.

**Definition 4** (Ancestral same-distribution condition (ASDC)). *In the interaction network $G^*$ a balanced interaction model, generic variable $W$ to satisfies the ancestral same-distribution condition (ASDC) if for all unit $i$, 1) $Pa(W_i)$ satisfies ASDC, and 2) $Pa(W_i) \subseteq \mathcal{V}_{(i)}$, and 3) for any different unit $j \neq i$, $Pa(W_i)$ and $Pa(W_j)$ have the same set of generic variables, and their exogenous errors $U_{W_i}$ and $U_{W_j}$ have the same distribution. (When i=j, the condition is automatically satisfied.)*

For example, in Figures 4(a) and 4(b), $X$ satisfies ASDC in the former (assuming the condition on exogenous errors is satisfied) but not in the latter, since in the latter $Pa(X_1) \neq Pa(X_2)$. ASDC implies IID as stated in the following lemma.

**Lemma 1.** *If $W$ satisfies ASDC, then any two explicit variables $W_i$ and $W_j$ are IID (Independent and Identically Distributed.)*

**Remark 2.** *The descendants of an ASDC variable need not be IID. For example, in Figure 4(a), $X$ satisfies ASDC, and $Y_i$ and $Y_j$ are descendants of $X_i$ and $X_j$. $Y_i$ and $Y_j$ have different sets of parents, making their distributions different, so $Y$ is non-IID.*

### 3.3 Quantity of Interest: True Average Causal Effect (TACE)

We generalize traditional ACE to the non-IID setting. Examine the interactions depicted in Figure 5(c). Unit $i$'s treatment $X_i$ affects their outcome through unit $j$'s outcome $Y_j$. $X_i \rightarrow Y_j \rightarrow Y_i$ is a "spurious" causal path. We are interested in computing the ACE of a unit's treatment on their outcome, excluding the effects transmitted via spurious paths from their neighbors/contacts. In an experimental setting, interactions can be eliminated by isolating all subjects. In an observational setting, where we are given the non-IID data, we are interested in computing the average causal effect of treatment on outcome *as if all units were isolated*. We present the formal definition below.

**Definition 5** (True Average Causal Effect ($TACE_{XY}$)). *Let $M^*$ be an interaction model. True average causal effect of $X$ on $Y$, denoted as $TACE_{XY}$, is defined as the ACE of $X$ on $Y$ in the isolated interaction model $IM^*$ corresponding to $M^*$.*

$TACE$ is the non-IID version of $ACE$ and is the same as $ACE$ in a traditional causal model where all samples are isolated. Again, without loss of generality, we assume the difference between the treatment value and the outcome value is 1, i.e. treatment is $X = c + 1$ and the outcome is $X = c$.

## 4 Defining, Quantifying, Detecting and Removing Interaction Bias for TACE

Almost all machine learning algorithms including those that employ causal techniques assume that data are IID (Schölkopf [2022], section 3). In other words, the theoretical and performance guarantees of these algorithms are based on data being IID. As such it would be useful to determine conditions under which an algorithm meant for IID data can be applied on non-IID data with the certainty that the resulting *bias* would be negligible. We formally define interaction bias below.

**Definition 6** (Interaction bias). *Let balanced model $M^*$ be the true model that generated the (available) non-IID dataset $D$. Let $Q$ denote the query of interest and let $Q^*$ be its true value. Let $A$ denote an algorithm that outputs an unbiased estimate of $Q$ given data that are IID and the causal graph that generated the IID data. Let $G^\dagger$ denote an approximate causal graph constructed under the assumption that $D$ is IID such that no assumption in $G^\dagger$ is refuted by $D$. Let $\hat{Q}$ be the estimate computed by $A$ using $G^\dagger$ and $D$ as input. Interaction bias is given by $||Q^* - \hat{Q}||$.*

### 4.1 Quantifying Bias

We define the two main types of problematic graphical structures in a linear interaction network that introduces bias in the estimation of $TACE$.

**Definition 7** (Deflecting bias structure). *A deflecting bias structure for $TACE_{XY}$ in an interaction network $G^*$ is an open path between $X_j$ and $Y_i$ for $i \neq j$.*

Deflecting bias structures are open paths from one unit to another unit. For example, Figures 5(a) and 5(b) contain deflecting bias structures. The interaction network in Figure 5(a) has a directed open path between $X_j$ and $Y_i$, and the interaction network in Figure 5(b) has a confounded open path between $X_j$ and $Y_i$.

**Definition 8** (Reflecting bias structure). *A reflecting bias structure for $TACE_{XY}$ in an interaction network $G^*$ is an open path between $X_i$ and $Y_i$ through some explicit variable $W_j$ with $i \neq j$.*

Reflecting bias structures are open paths that go from a unit through another unit and back to the same unit. For example, Figures 5(c) and 5(d) contain a reflecting bias structure. In each of them,

there is an open path from $X_i$ to $Y_i$ through $Y_j$. In some cases, there can be a deflecting bias structure embedded in a reflecting bias structure, as in Figures 5(c) and 5(d). However, this is not necessary. Figure 5(e) contains only a reflecting bias structure ($X_i \rightarrow C_j \rightarrow Y_i$) but no deflecting bias structure.

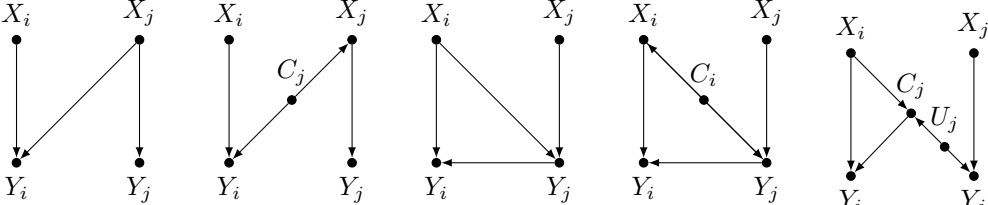

**(a)** Deflecting bias: $X_j$ causes $Y_i$ through a path. **(b)** Deflecting bias: $X_j$ and $Y_i$ are confounded. **(c)** Reflecting bias: $X_i$ causes $Y_i$ through $W_j (= Y_j$ here). **(d)** Reflecting bias: $X_i$ and $Y_i$ have a confounding path. **(e)** Reflecting bias only.

**Figure 5:** Two main types of interaction bias.

**Theorem 1.** *Let $M^*(G^*, S^*)$ be a balanced interaction model in which treatment variable $X_i$ and outcome variable $Y_i$ are not confounded by any variable in $\mathcal{V}_i$, $\forall i$. Let $D$ be the available data generated by $M^*$ and let $G^\dagger$ be the approximate graph constructed using $D$. Let $TACE_{XY}$ be identifiable in $G^\dagger$ and be given by $\beta_{YX}$, the regression coefficient of $Y$ on $X$. Let $\alpha$ denote the true value of $TACE_{X,Y}$ in $M^*$. If $X$ satisfies ASDC then the interaction bias is given by,*

$$\left| E[\hat{\beta_{YX}}] - \alpha \right| = \left| \frac{1}{n} \sum_{1 \leq i \leq n} \sum_{p \in P[iji]} Val(p) \frac{\sigma^2_{R_p}}{\sigma^2_X} - \frac{1}{n(n-1)} \sum_{1 \leq i \leq n} \sum_{p \in P[ji]} Val(p) \frac{\sigma^2_{R_p}}{\sigma^2_X} \right|,$$

*where $P[iji]$ is the set of reflecting bias structures between $X_i$ and $Y_i$ through any explicit variable $W_j$ of unit $j$ with $i \neq j$, $P[ji]$ is the set of deflecting bias structures between $X_j$ and $Y_i$ with $i \neq j$, and $R_p$ is the root of path $p$.*

It follows from Theorem 1 that in a balanced interaction model in which no $X_i$ and $Y_i$ are confounded by any variable in $\mathcal{V}_i$, the reflecting and deflecting structures are the only two structures that will bias the identification of $TACE$. Note that although definition of interaction bias (Definition 6) on TACE is for any unbiased estimator for $ACE$, we focus only on the ordinary least squares estimator in this paper. This is because among the class of unbiased linear estimators, the OLS estimator has the minimum variance [Johnson et al., 2014].

We exemplify theorem 1.

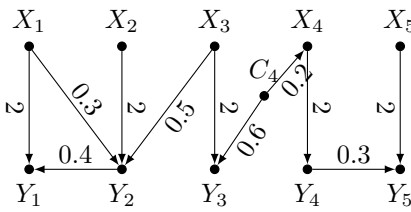

**Figure 6:** Interaction network with 4 units. The numbers represent edge coefficients. ($C_1, C_2, C_3, C_5$ are omitted)

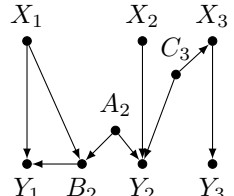

**Figure 7:** Interaction network with 3 units. (Other $A, B, C$ variables including $A_1, B_1, \ldots$ are omitted)

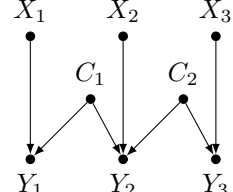

**Figure 8:** Interaction network with 3 units. ($C_3$ is omitted)

**Example 1.** *Figure 6 shows an example of an interaction model with 4 units where $X_1, \ldots, X_5$ are the treatments, and $Y_1, \ldots, Y_5$ the outcomes. The numbers on the edges are the edge coefficients. $C$ satisfies ASDC, and $C_i$ for $i = 1, 2, 3, 5$ are omitted from the graph for simplicity.*

*Suppose we want to estimate the $ACE$ of $X$ on $Y$ as if the units were isolated:* **Input:** *the interaction network $G^*$ as shown in Figure 6 (no parameter i.e., $S^*$ is not an input),* **Output:** *the $TACE_{XY}$ (should equal to 2). If we estimate $ACE_{XY}$ ignoring the connections between units, our estimator will be $\hat{\beta_{YX}}$, with $Y = \{Y_1, \ldots, Y_5\}$ and $X = \{X_1, \ldots, X_5\}$. This is because ignoring the connections,*

*the graph becomes $X_i \to Y_i$ separated for $i = 1, \ldots, 5$, so is essentially $X \to Y$ Pearl [2009]. However, by Theorem 1,*

$$|\beta_{YX} - 2| = |\frac{0.3 \cdot 0.4}{5} - \frac{1}{20} \cdot 0.5 - \frac{1}{20} \cdot 2 \cdot 0.4 - \frac{1}{20} \cdot 0.5 \cdot 0.4 - \frac{1}{20} \frac{0.6 \cdot 0.2\sigma_C^2}{\sigma_X^2} - \frac{1}{20} \cdot 2 \cdot 0.3|$$
$$\neq 0.$$

*Hence, the result is biased, and does not give us what we want. We show later in Theorem 2 how to compute an unbiased estimate of $TACE$.*

## 4.2 Detecting Bias

In this section, we provide a graphical criterion resulting from Theorem 1, to detect interaction bias.

**Corollary 1.** *Let $M^{**}(G^{**}, S)$ be a balanced interaction model in which $X$ satisfies ASDC and TACE is identified as $\beta_{YX} = \alpha$ in the approximate graph, then interaction bias exists iff $G^{**}$ contains a reflecting or deflecting bias structure.*

For example, Figure 7 contains both reflecting and deflecting bias structures. Figure 8 does not contain any bias structure. So Figure 7 has interaction bias and Figure 8 does not. Note that the interactions in Figure 8 do not qualify as bias structures by Definitions 7 and 8.

## 4.3 Removing Bias

Theorem 2 presents a technique for computing an unbiased estimate of TACE in cases where theorem 1 predicts significant bias. It proceeds by applying linear regression on a set of samples $B$ that satisfy the condition that no bias inducing structures exist between any two distinct units $i$ and $j$. In particular, a subset of samples/units $B$ is termed as a **bias-free subset** for $TACE_{XY}$ if no reflecting bias structures exist for any $i \in S$ and no deflecting bias structures exist in $G_S^*$ where $G_S^*$ is the latent projection of $G^*$ on $B$ (Definition 2.6.1, Pearl [2009]). For example in figure 6, $B$ comprises of units 2 & 5 and $G_S^*$ is $X2 \to Y2$ $X_5 \to Y_5$. However, $B$ *is not unique* for a given interaction network. Another candidate for $B$ is units 2 & 4 and the associated $G_S^*$ is $X2 \to Y2$ $X_4 \to Y_4$. An algorithm for constructing $B$ is presented in Algorithm 1, with an example and discussion in the appendix. This algorithm starts by randomly initializing $B$ with a sample. Then it goes through the rest of the samples and adds a sample to $B$ if its inclusion does not create bias structures in the resultant graph, $G_S^*$.

---

**Algorithm 1** Select a bias-free subset $B$ from an interaction network $G^*$ and return the largest subset from $t$ iterations

---
    **Input:** an interaction network $G^*$, iterations $t$
    **Output:** the largest bias-free subset $B$ selected from $t$ iterations
1: **function** FINDSUB($G^*$, $t$)
2:     $\mathbf{B} = \emptyset$
3:     **for** $i = 1, \ldots, t$ **do**
4:         $Units =$ randomly sorted list $1, \ldots, n$
5:         $B = \{Units[1]\}$ (The indices for $Units$ start from 1)
6:         **for** $i = 2, \ldots, n$ **do**
7:             **if** $Units[i]$ has no reflecting bias structure in $G^*$ **then**
8:                 **if** $Units[i]$ has no deflecting bias structure in $G^*$ with an element in $B$ **then**
9:                     $B = B \cup \{Unit[i]\}$
10:         $\mathbf{B} = \mathbf{B} \cup \{B\}$
11:     **return** Largest $B$ in $\mathbf{B}$

---

**Theorem 2.** *Let $G^*$ be an interaction network. Given the conditions in Theorem 1 and 'B' a bias-free subset for $G^*$, $TACE_{XY} = E[\hat{\beta_{YX}}]$ where the regression coefficient is calculated using only samples in set $B$.*

Note that bias-free subset of samples $B$ *used in Theorem 2 is not always IID*. While we insist that no reflecting or deflecting bias structures exist in $G_S^*$, we do not restrict other forms of interactions

among these samples. For example, in Figure 8, Units {1, 2, 3} constitute a bias-free subset. In this case, $Y$ is not IID ($Y_1$ and $Y_2$ are dependent, $Y_2$ and $Y_3$ are dependent) and hence the bias-free subset is non-IID.

Also note that to compute an unbiased estimate using Theorem 2, we have at our disposal a smaller set of samples; so the variance of estimation will be larger. There is a trade off between ignoring interaction (large bias, small variance), and using theorem 2 (no bias, large variance). It remains future work to quantify the variance of the estimator in Theorem 2 for different interaction models, but in Section 5, we provide simulation results and case analysis study to empirically show its performance.

**Applicability of theorems 1 & 2 to real world problems:** A natural question that arises at this juncture is whether we need an entire interaction network to apply these results to real world problems. Theorem 1 quantifies bias and in doing so reveals to us if and how various factors such as sample size and strength of connections (value of path coefficients) influence bias. This in turn allows us to use available information about the problem from prior experience, domain knowledge or external sources to determine if bias would be negligible or not. Specifically, bias becomes smaller as the number of bias-structure-free samples increases. In fact, if the numbers of deflecting and reflecting structures are fixed, the bias terms diminishes as $n$ increases, indicated by the $1/n$ for the reflecting bias term and $1/n(n-1)$ for the deflecting bias term. It is also evident that if the values of path coefficients are high, $Val(p)$ would be high and this will result in increased bias. Finally, if the interaction connections are sparse (fewer edges between units), the reduction in the total number of paths could potentially lower bias but more importantly the number of samples in the bias-free set $B$ used in theorem 2 will tend to be larger, which in turn will help in computing better quality estimates.

# 5 Experiments

## 5.1 Simulations

**Simulated Model** We randomly generate balanced interaction network with $n$ units (i.e., the sample size is $n$), with $C_i \to X_i \to Y_i$ and $X_i \to M_i$ for all $i = 1, \ldots, n$. For all ordered pairs of distinct units $i, j$, we randomly add deflecting bias structures in the form of $X_i \leftarrow C_i \to Y_j$ with probability $dRate$. For all units $i$, we randomly add reflecting bias structures in the form of $X_i \to M_k \to Y_i$ with probability $rRate$ for a random $k \neq i$.

**Experiment: Bias of REG** It follows from theorem 1 that larger sample sizes and smaller path values on the bias structures result in smaller bias. We perform two simulations to show how bias varies as a function of sample size and path values. We simulate data such that for each variable, the exogenous error term follows a Gaussian distribution with mean 0 and standard deviation 1. For each set of parameters, we randomly generate an interaction network, and simulate the data 10000 times. Each time, we record the result from a naive regression of $Y$ on $X$ (REG). As a comparison, we also record the result from Theorem 2 (THM-2). We run the algorithm (provided in the appendix) to randomly select bias-free subsets for 10 times and select the largest subset.

**Simulation 1:** $X_i \to Y_i$'s edge coefficient is 100, the edge coefficients of $C_i \to X_i, X_i \to M_j, M_j \to Y_i$ are all set to 10, the numbers of deflecting bias structures and additional reflecting bias structures are both 100.

**Simulation 2:** Number of units $n = 1000$, $X_i \to Y_i$'s edge coefficient is 100, the numbers of deflecting bias structures and additional reflecting bias structures are both 100. The results are plotted in Figure 9. As seen in the plots, as $n$ increases or the path values on the bias structures decreases (both with all other parameters fixed), $\beta_{YX}$ from a naive regression approaches $TACE$. Such results coincide with Theorem 1. The $\beta_{YX}$ computed by THM-2 is very close to TACE and the two lines almost overlap.

## 5.2 Case Study

**Settings** We are interested in analyzing the effect of tutoring time on students' grades. In particular, we wish to compute the effect provided through the tutoring program only, but not through "side effects" from other units, such as learning from classmates, although such interactions are encouraged in this scenario. For instance, unit $i$ might help unit $j$ understand the course materials better which in turn might improve $j$'s grade. If unit $i$ helped unit $j$ improve their understanding and unit $j$ states this in the peer review, then it would boost $i$'s grade. To construct an interaction network and apply

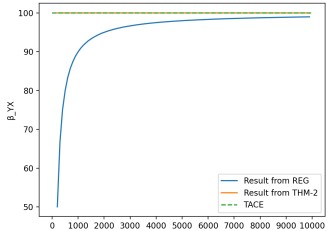 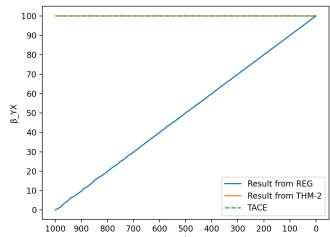

**Figure 9:** Left: $\beta_{YX}$ vs. number of units $n$. Right: $\beta_{YX}$ vs. path value on the bias structures. $TACE = 100$.

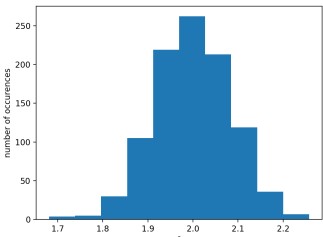 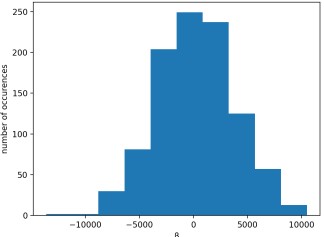

**Figure 10:** Left: estimated $TACE$ distribution from THM-2. Right: estimated $TACE$ distribution from REG.

our results, we ask the students to fill out a survey including 1) their tutoring time, 2) their grade, 3) whom they helped, 4) who helped them, 5) peer review score.

**Construction of the Interaction Network** Three generic variables are $T$ (tutoring time in hours), $U$ (understanding of course materials), and $R$ (grade). For each unit $i$, $T_i \to U_i \to R_i$. In addition, if $i$ helped $j$, add $U_i \to U_j$ (deflecting bias structure). If $i$ first helped $j$ and $j$ mentioned this in the peer review and thus boosted $i$'s grade, add $U_i \to U_j \to R_i$ (reflecting bias structure). We assume no additional back-and-forth help happens.

**Simulation** Let there be 500 students, assume each student on average help 5 other students, and the other student has a 0.5 chance of helping back. Let $TACE = 2$, and the $U_i \to U_j$ and $U_i \to R_j$ edges both have the value 2. We randomly generate an interaction network and simulate data based on these parameters.

**Results** We apply THM-2 to select a bias-free subset, and compute $\beta_{GT}$ using data from that subset. We get the result 1.963, with the size of the subset 72. As a result, the effect of tutoring time on students' grades not through other units is estimated to be 1.963, which is close to the ground truth $TACE$ (2). We further repeat the experiment 1000 times to show the distribution of the results. Each time a random structure is generated and random data are simulated. THM-2 is on average able to select a bias-free subset of size 76, and the average recovered $TACE = 2.0002$. The result from *REG* had a *significantly high bias with TACE averaging at* 194.11. Also since every time the data are regenerated, the model is different, and REG uses all the data, it has a larger variance. The two plots in Figure 10 show the distribution of results from THM-2 and REG. The histograms of the results of $\beta_{YX}$ computed by THM-2 and REG are shown in Figure 10.

## 6 Related Work

One of the most studied concepts related to interactions among units is interference [Cox, 1958]. Majority of literature in empirical fields assume no-interference. In fact, SUTVA is a common assumption in causal inference [Rubin, 1978]. Recent years have witnessed a rise in papers on interference that employ graphical models. These include Ogburn and VanderWeele [2014] that was the first to model interference using DAGs, Sherman and Shpitser [2018] that modeled interference using chain graphs which permits modeling unknown interactions between units and Bhattacharya et al. [2020]that proposed structure learning methods for chain graphs. These works rely on partial interference which divides units into equal-sized blocks under the assumption that interactions occur

only within a block but not across different blocks. [Nabi et al., 2020] developed methods for identification and estimation of multiple queries under conditions of interference and homophily, and applied the results to the problem of ad-placements.Sobel [2006] was the first to notice the effect of interference in the housing mobility problem, and proposes causal estimands for this application.

Aronow and Samii [2017], Sussman and Airoldi [2017] modeled general interference (without assuming partial inference) by constructing a function to define a unit's exposure level on the number of treated neighbors they have. The methods are less restricted than partial interference methods, and allow units to be affected by any number of neighbors. However, they are limited to interference and do not handle other forms of interactions.

Jagadeesan et al. [2020] proposed a quasi-coloring method to estimate direct effect under interference using experimental data. However, it does not easily generalize to observational studies. Other papers along a similar direction include Fatemi and Zheleva [2020], which proposed experiment design to minimize interaction bias and selection bias at the same time, and Liu and Hudgens [2014], which proposed a two-stage randomization design to minimize interference bias. Tchetgen Tchetgen et al. [2021] proposed a g-computation method, which is the first to model general interference using graphical models (chain-graphs), but requires the interference effects to be symmetrical between units. Sävje et al. [2021] and Hudgens and Halloran [2008] defined queries similar to TACE, named EATE and PADE, respectively. These queries generalize traditional ACE to allow a unit's outcome to be affected by treatments of other units. However, they do not allow outcomes to be affected by other units' variables other than treatments.

Hudgens and Halloran [2008] defined six types of queries in the problems involving interference. Work in interference that focuses on different queries/problems include a few as follows. VanderWeele et al. [2012b] is the first to decompose the spillover effect (the effect of a unit's treatment on another's outcome (Quammen [2012])) to contagion and infectiousness effects using counterfactual mediation analysis. Shpitser et al. [2017] presented decomposition for units with unknown and symmetrical interaction patterns and analyzed different interference paths. In linear models, the contagion and infectiousness effects reduce to the directed paths from $X_j$ to $Y_i$. Moreover, their work does not handle reflection bias. Hu et al. [2021] was the first to define and provide estimands for the average indirect effect. VanderWeele et al. [2014] developed methods for sensitivity analysis under interference.

Other types of interactions include the contagion effects, which are defined as a unit's outcome affecting another unit's outcome [VanderWeele and An, 2013]. Work on this line usually used longitudinal data, including Burt [1987], Lyons [2011], VanderWeele et al. [2012a]. Homophily effects are defined as the behavior of connected units are similar [Jagadeesan et al., 2020]. Work in this line include McPherson et al. [2001], Jagadeesan et al. [2020]. The existing work above does not model interactions using graphical models.

# 7   Conclusions

In this paper, we represent interactions among units using causal graphical models. We derive theorems to quantify the interaction bias for average treatment effects in linear models. We provide sufficient and necessary graphical conditions to detect interaction bias. Additionally, we develop a method to compute an unbiased estimate of causal effect in cases where blindly assuming IID is expected to yield a significant bias. Finally, we discuss the performance of our method through simulation studies.

## Acknowledgments and Disclosure of Funding

We would like to thank anonymous reviewers for their comments, and Totte Harinen and Rumen Iliev for helpful discussions. This research was supported in parts by grants from the National Science Foundation [#IIS-2106908 and #2231798], Office of Naval Research [#N00014-21-1-2351], and Toyota Research Institute of North America [#PO-000897].

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
