# OpenReview forum: "Causal Inference with Non-IID Data using Linear Graphical Models"
_NeurIPS.cc/2022/Conference — NeurIPS 2022 Accept_

### Official Review · Reviewer_bAan · 2022-07-07

**Rating:** 7
**Confidence:** 2
**Soundness:** 3 good
**Presentation:** 2 fair
**Contribution:** 3 good

**Summary:**

The goal of the paper is to study non typical forms of interference that cause iid violation and bias, and how these affect Average Causal Treatment Effects estimates. Moreover the authors propose set of conditions which when they hold true, violation of iid assumption when using an algorithm meant for iid will result in negligible bias of the estimated ATE.


**Questions:**


While the paper's ultimate goal seems very relevant for real datasets, i would like to ask the authors some questions about some points:
1. In Figure 5 I do not understand why figures 5c-5e do not contain a deflecting path as well. According to the open path definition (no collider), in all these graphs there is
an open path from X_j to Y_i.

2. Does the proposed method assume that the user almost needs to know the full graph, and if so, don't we then already know if there is interference or not?

3. Could you please clarify what is meant by "no parameter i.e., S* is not an input" in line 207?

**Limitations:**

No limitations are discussed by the authors. I do not see any limitation with respect to societal impact for this work.

**Strengths And Weaknesses:**

Strengths:

The paper is well written  and with gradual introduction of the terminology needed for the reader. Also it addresses a very interesting statistical problem, relevant for many fields.

Weaknesses:

The interaction bias formula contains the variance of the root path node R_p. This means that the authors assume that R_p must be known in a real dataset.
I would like the authors to elaborate how realistic is this requirement, given that in many use cases the main problem is to identify the root cause of a target.

I would like to see application on some real-dataset if possible. Also the comparison with Regression in the simulated graph is the absolute minimum. Could the authors include comparisons of their method with some state of the art ATE method?

Some comments about structure:
- Captions of Figure 5, 6 and 7 cannot be read, as they are very close to each other. Please, add appropriate spacing.
- line 307, "some polots above' not 'below'.
- Minor typos: drop 'to' line 134, l.115 is -> are

---

> ### Author Response · Authors · 2022-08-02
> **Response to bAan**
>
> Assuming $R_p$ is known concern: $R_p$ is known if the full causal structure is known and knowledge of the full structure is assumed by most work on causal identification (Pearl 2009). Given the causal graph, we can simply trace the paths in the causal structure to locate the root. In the case where the root is unobserved or the structure is incomplete, one will need structure learning methods or domain knowledge to fill in the gap, which is beyond the scope of this paper.
>
> State of the art concern: The state of the art ATE methods assume IID data and, indeed, our main results quantify the bias induced when iid is blindly assumed. To the best of our knowledge, our work is the first to quantify such bias. The few other works on interference, briefly discussed in lines 316-318, deal with partial interference -- an unrealistic assumption. For example, disease spreading is not limited to families unless all families lead an isolated life; a waiter at a restaurant or an uber driver can also spread the disease beyond family boundaries. Most other popular techniques (generally discussed in section 6) only deal with $X_i\rightarrow Y_j$ type of interactions but not with general interactions as assumed in our paper.
>
> Figure 5 concern: 5c and 5d have deflecting paths and we mentioned that in the paragraph right above Figure 5. Thank you for noticing the error that 5e also contains a deflecting path. We will update 5e by removing $Y_j\rightarrow C_j$ and adding $Y_j\leftarrow D_j\rightarrow C_j$, which will become a correct example for a model with reflecting bias but no deflecting bias.
>
> Interference obvious from graph concern: We assume that the interaction graph is given. We note that in many settings the rudimentary information needed to construct interaction graphs is often available. For example, in many countries during the pandemic contact tracing was meticulously conducted and as such it is known who interacted with whom. In social media such as twitter and facebook, the interactions among units are stored and exploited. Of course, these are sensitive data, closely guarded and not easily available for use, without iron-clad NDAs.
>
> You are right in saying that just by looking at the interaction graph one can decide if there is interference or not. We point out that in literature interference is identified as a directed path from $X_i$ to $Y_j$. Traditional interference thus corresponds to a special case of what we call the deflection bias. But interference is not the only source of bias. Other types of deflecting bias and reflecting bias structures can also cause bias. This is the first work that identified reflection bias structure and proved that reflecting and deflecting bias structures are the only structures that can induce bias while estimating causal effect. Thus $X_i\leftarrow C_i\rightarrow Y_j$ creates interaction bias, but $Y_i\leftarrow C_i\rightarrow Y_j$ does not. Thus, prior to this work it was only possible to know by inspecting the graph whether or not bias due to interference (deflecting bias structure) is to be expected. Now by inspecting the interaction graph we can immediately conclude whether or not any bias is to be expected.
>
> "no parameter i.e., $S^*$ is not an input" in line 207 question: We meant the structural equations (the parameters) are not given. Please see Definition 1 for the definition of $S^*$. We will clarify that in the revision. This is a common setting in causal identification literature, where those parameters are usually the goal of identification. We wanted to show through Example 1 that there is interaction bias if IID is wrongly assumed, and we show how to debias in Theorem 2 (without the parameters given to us).

---

> > ### Comment · Reviewer_bAan · 2022-08-05
> > **response to authors' response**
> >
> > Thank you for your responses and for adjusting your paper accordingly to my comments. On my first comment i was biased from a causal discovery point of view, as there the biggest challenge is actually to recover the graph. Nevertheless, I completely acknowledge the contribution of your work, for the cases where the graph is known. I would like to ask you to add your second to last paragraph in your answer, to your Discussion, as it clearly demonstrates the contribution of your paper.

---

> > > ### Author Response · Authors · 2022-08-08
> > > **Response**
> > >
> > > Thank you for your reply. We agree and will add the paragraph to the revised paper.

---

### Official Review · Reviewer_wxC8 · 2022-07-12

**Rating:** 6
**Confidence:** 3
**Soundness:** 3 good
**Presentation:** 2 fair
**Contribution:** 4 excellent

**Summary:**

This paper studies causal inference from non-iid data when there are interactions among units, in the case of linear causal relationships. In particular, it analyzes different types of interactions, detects and quantifies the corresponding interaction bias, and derives conditions under which it is safe to ignore interactions and estimate the causal effect with methods that are designed for iid data.

**Questions:**

I would like to suggest the authors improve their writing. For experimental studies, it would be more convincing if the authors could consider different types of interactions that are considered in previous sections.

**Limitations:**

The authors did not discuss the limitations and potential negative societal impact of their work

**Strengths And Weaknesses:**


Strengths:
The studied problem is very interesting and meaningful because, in many real-world problems, the iid assumption is hard to hold. As far as I know, this work is original.

The authors show theoretical results on detecting and quantifying interaction bias, as well as a technique for removing the bias.

Weakness:
This paper does not seem easy to follow; there are too many definitions, and the writing also needs improvement.

For the notation Xi, sometimes i is written as a subscript, but sometimes it is not.

The experimental studies do not seem enough.

---

> ### Author Response · Authors · 2022-08-02
> **Response to wxC8**
>
> Too many definitions concern: Please see the response to xxrS.
>
> Interactions of different types in the experiments concern: In the experiments, we presented an analysis when bias inducing structures are present. In the first two simulations (section 5.1), we studied how bias varies with changes in sub parameters in the presence of deflecting and reflecting bias. We included different types of interactions (deflecting and reflecting bias) in those simulations. Same with the simulation in the appendix. In the third simulation (section 5.2) also, we included both deflecting and reflecting bias structures. We will highlight that in the revised paper.
>
> Regarding the limitations and potential negative societal impact: This is the first work that analyzes the bias of blindly applying IID methods on non-IID data and can be extended in several ways such as quantifying the variance of the unbiased estimand (line 240-244) and handling counfoundedness from the same unit (line 188). We intend to address these in our future work and will highlight these in the revision. This work has no negative societal impact. On the contrary, our line of research alerts to the perils of blindly making IID assumptions.

---

> > ### Comment · Reviewer_wxC8 · 2022-08-09
> > **Thank you**
> >
> > Thank you for the response. I will keep my score "weak accept". Hope the presentation can be further improved.

---

### Official Review · Reviewer_xxrS · 2022-07-14

**Rating:** 3
**Confidence:** 1
**Soundness:** 2 fair
**Presentation:** 1 poor
**Contribution:** 2 fair

**Summary:**

The paper studies the causal inference problem with interference under a linear model. It assumes known interference structure, and present conditions under which the bias due to ignoring interference is negligible. It also presents a linear regression method on a set of samples that satisfies the condition that no bias-inducing structures exist between any two distinct units.


**Questions:**

Is it possible to illustrate the main idea, and under what conditions you can identify/remove the bias, without resorting to these new terminologies you introduce?


**Limitations:**

The linearity assumptions seem rather strong and do they ever hold in practice (for the whole network)?

Also the same for the other assumptions. More discussions on when one may be able to detect and remove the bias in a real context would be very helpful.

**Strengths And Weaknesses:**

I am not aware of many previous papers tackling this problem using the language/perspective that the authors use, so that seems something novel. On the other hand, my feeling is that the authors introduce too many terminologies so that the main idea is buried behind these notations.

The authors motivate the discussion with an example of COVID-19 but finish with a simulation (!) on student grades. Why not make it consistent? This is a simulation anyway.

---

> ### Author Response · Authors · 2022-08-02
> **Response to xxrS**
>
> Student grade simulation concern: We chose the student grade setting for two reasons; first, the setting is likely to be familiar to any person reading this paper and second, it is a simple model in which we could jointly exemplify both reflecting and deflecting biases and thus communicate various nuances to the reader. A realistic COVID-19 simulation involves modeling a rather complex setting (with numerous variables and interactions) and introducing medical and epidemiological terminologies that would make comprehension harder for the reader.
>
> New terminologies concern: The main ideas devoid of mathematical definitions are as follows,
> 1. Blindly assuming that data are IID when in fact they are non-IID can potentially bias the outcome of a research study.
> 2. Such bias can occur for the query: causal effect of treatment on outcome, when there is an open (not necessarily directed) path from the treatment of unit i (a) to the outcome of unit $j$ $(j\neq i)$ and/or (b) to the outcome of unit i itself such that an intermediate node on the path belongs to unit $j$ $(j\neq i)$
>
> 3. The formula/estimand in theorem 1 quantifies the bias. Furthermore, only the above two types of interaction structures can induce bias.
>
> 4. In the presence of such bias inducing structures it is still possible to compute an unbiased estimate by selecting a subset of samples S such that no biasing paths (2(a) & 2(b)) exist in the interaction graph corresponding to samples in $S$.
>
> 5. Most importantly, such a debiasing procedure does not require the selection of IID samples i.e. $S$ may contain interactions between them such as ($Y_i\leftarrow C_i\rightarrow Y_j$) (line 238). Such a debiasing procedure can also be done in polynomial time (Appendix A).
>
> 6. In the simulations we randomly generate interaction models and show that the bias can be huge if IID is wrongly assumed on non-IID data. The debiasing method in this paper yields an unbiased estimate. We further show that, as the number of IID samples increases, and as the strength of bias structure decreases, the overall interaction bias decreases.
>
> We shall include a summary paragraph such as the one above in our paper. However, we note that mathematically proving 1 to 4 is non-trivial and requires introduction of the various terminologies (by terminologies we assume you meant definitions 1-8). To make comprehension easy we have provided examples after each definition and will provide additional examples in the appendix.
>
> Linearity assumption concern. We would like to answer your concern as follows. It is hard to think of a principle or a method in causal analysis, perhaps even in science more generally, that was not first conceptualized, quantified, and exemplified in linear models before it was understood and generalized to non-linear or nonparametric models [1]. The entire enterprise of graphical models, for example, was first conceived [2] in linear path-diagrams, 60 years before it was generalized to non-parametric models. Heckman's seminal work on sample selection bias [3], as another example, was demonstrated on linear models in 1979, 35 years before more general methods were developed. We see no theoretical impediments to having such generalizations emerge for the principles and methods exemplified in our paper, though this would take us beyond the scope of the current paper, whose main focus is the development of a mathematical framework for analyzing the highly neglected problem area of non-IID data.
> Moreover, from a practical viewpoint, linearity assumptions dominate the practice of statistics. Linear regression, for example, is one of the simplest and most common machine learning algorithms [4], and applied successfully to real-world problems such as crime rate, geographic dynamics, body fat and health care. See references [5, 6, 7].
>
> [1] Pearl, J. Linear models: A useful “microscope” for causal analysis.”
>
> [2] Wright, S. (1921). Correlation and causation
>
> [3] Heckman, James J. Sample Selection Bias as a Specification Error.
>
> [4] Kelter, R. Bayesian identification of structural coefficients in causal models and the causal false-positive risk of confounders and colliders in linear Markovian models
>
> [5] Zheng Yuan & Yuhong Yang (2005) Combining Linear Regression Models
>
> [6] Lindsey C, Sheather S. Variable Selection in Linear Regression
>
> [7] Riu, Jordi, and F. Xavier Rius. Assessing the accuracy of analytical methods using linear regression with errors in both axes.
>
> Other assumptions concern: the symmetrical assumptions in this paper (interaction models are balanced, certain variables satisfy ASDC, etc) are weaker than traditional causal inference work which assumes IID. I.e., when IID is assumed, the interaction models must be balanced, and all variables satisfy ASDC. But when an interaction model is balanced, there could still be interactions to make the variables non-IID and hence IID is a stronger assumption.

---

> > ### Comment · Reviewer_xxrS · 2022-08-08
> > **Thank you for the detailed response**
> >
> > The author does partially address my concern about linearity, so I would be happy to raise my rating to 4.

---

> > > ### Author Response · Authors · 2022-08-08
> > > **Response**
> > >
> > > Thank you for your reply.

---

### Author Response · Authors · 2022-08-02
**Author response to the all reviewers**

We would like to thank all the reviewers for taking their time to read our papers and provide helpful reviews. We are delighted to see that the novelty and importance of this work is acknowledged.

We first answer a general concern regarding real-world context/data by xxrS and bAan. In methodology-centered research, the advantage of demonstrating ideas using simulated data lies in the availability of ground truth, against which one can calibrate the performance of proposed techniques. In our specific case, the asymptotic bias induced by the data-generating process can be computed analytically, and can be used to gauge the performance of the techniques we proposed, when applied to finite samples. No such gauge would be available in real world data, thus preventing us from distinguishing methodological weaknesses from data idiosyncrasies.

We will respond to individual questions and concerns in the following messages, and we welcome further discussions.

---

### Meta-Review · Area_Chair_cgqF · 2022-08-30

**Recommendation:** Accept
**Confidence:** Less certain

**Metareview:**

This manuscript offers some potentially important theoretical and practical contributions in the area of causal inference, particularly surrounding the assumption of iid data and how this assumption can be violated in the presence of interference. The manuscript describes a modeling framework for such interactions and derives some theoretical analysis regarding the detection such interactions and when they may be ignored (and the data nonetheless treated as iid). For a field experiencing considerable growth and momentum, these developments are timely and have a strong potential for impact.

Although there was some disagreement among the reviewers, the balance of opinion was in favor of acceptance, especially after the insightful author rebuttals and ensuing discussion between authors and reviewers. I recommend that the authors take the fruits of these discussions into account when preparing an updated version of this manuscript.

**Award:**

No

---

### Decision · Program_Chairs · 2022-09-14

Accept